# Alterations in Homologous Recombination-Related Genes and Distinct Platinum Response in Metastatic Triple-Negative Breast Cancers: A Subgroup Analysis of the ProfiLER-01 Trial

**DOI:** 10.3390/jpm12101595

**Published:** 2022-09-27

**Authors:** Elise Bonnet, Véronique Haddad, Stanislas Quesada, Kim-Arthur Baffert, Audrey Lardy-Cléaud, Isabelle Treilleux, Daniel Pissaloux, Valéry Attignon, Qing Wang, Adrien Buisson, Pierre-Etienne Heudel, Thomas Bachelot, Armelle Dufresne, Lauriane Eberst, Philippe Toussaint, Valérie Bonadona, Christine Lasset, Alain Viari, Emilie Sohier, Sandrine Paindavoine, Valérie Combaret, David Pérol, Isabelle Ray-Coquard, Jean-Yves Blay, Olivier Trédan

**Affiliations:** 1Department of Medical Oncology, Centre Léon Bérard, 69008 Lyon, France; 2Department of Biopathology, Centre Léon Bérard, 69008 Lyon, France; 3Institut Régional du Cancer de Montpellier (ICM), 34298 Montpellier, France; 4Centre Hospitalo-Universitaire Dupuytren, 87042 Limoges, France; 5Biostatistic Unit, DRCI, Centre Léon Bérard, 69008 Lyon, France; 6CNRS UMR 5286, INSERM U1052, Université Claude Bernard Lyon 1, 69008 Lyon, France; 7Cancer Research Center of Lyon (CRCL), 69008 Lyon, France; 8Institut de Cancérologie de Strasbourg Europe, 67200 Strasbourg, France; 9CNRS UMR 5558, Laboratoire de Biométrie et Biologie Évolutive, Université Claude Bernard, 69622 Lyon, France; 10Department of Prevention and Public Healthcare, Centre Léon Bérard, 69008 Lyon, France; 11Synergie Lyon Cancer, Bio-Informatics Platform, Centre Léon Bérard, 69008 Lyon, France; 12Faculté de Médecine Lyon Est, Université Claude Bernard Lyon 1, 69373 Lyon, France; 13UNICANCER—Fédération des Centres de Lutte Contre le Cancer, 75013 Paris, France

**Keywords:** triple-negative breast cancer, TNBC, platinum-based chemotherapy, DNA repair, homologous recombination, HRR genes, *BRCA*

## Abstract

Background: a specific subset of metastatic triple-negative breast cancers (mTNBC) is characterized by homologous recombination deficiency (HRD), leading to enhanced sensitivity to platinum-based chemotherapy. Apart from mutations in *BRCA1/2* genes, the evaluation of other HRD-related alterations has been limited to date. As such, we analyzed data from mTNBC patients enrolled in the ProfiLER-01 study to determine the prevalence of alterations in homologous recombination-related (HRR) genes and their association with platinum sensitivity. Methods: next-generation sequencing and promoter methylation of *BRCA1* and *RAD51C* were performed on tumors from patients with mTNBC, using a panel of 19 HRR genes. Tumors were separated into three groups based on their molecular status: mutations in *BRCA1/2*, mutations in other HRR genes (*BRCA1/2* excluded) or *BRCA1/RAD51C* promoter methylation and the absence of molecular alterations in HRR genes (groups A, B and C, respectively). Sensitivity to platinum-based chemotherapy was evaluated through the radiological response. Results: mutations in *BRCA1/2* were detected in seven (13.5%) patients, while alterations in other HRR genes or hypermethylation in *BRCA1* or *RAD51C* were reported in 16 (30.7%) patients; furthermore, no alteration was found in the majority of patients (*n* = 29; 55.8%). Among 27 patients who received platinum-based chemotherapy, the disease control rate was 80%, 55% and 18% (groups A, B and C, respectively; *p* = 0.049). Regarding group B, patients with disease control exhibited mutations in *FANCL*, *FANCA* and the *RAD51D* genes or *RAD51C* methylation; Conclusion: mutations in HRR genes and epimutations in *RAD51C* were associated with disease control through platinum-based chemotherapy. As such, apart from well-characterized alterations in *BRCA1/2*, a more comprehensive evaluation of HRD should be considered in order to enlarge the selection of patients with mTNBC that could benefit from platinum-based chemotherapy.

## 1. Introduction

Breast cancer (BC) represents the leading cause of death among females in the world [1]. A specific subset of BC, triple-negative breast cancer (TNBC), is characterized by the lack of estrogen and progesterone receptors and human epidermal growth factor receptor 2 (HER2) amplification. TNBCs exhibit specific features such as higher prevalence in younger patients, high histologic grade and more aggressive disease than other BC subtypes, leading to a poorer clinical outcome [2]. Indeed, over 30% of patients with TNBC will relapse with metastatic disease within three years; furthermore, the median overall survival (OS) of metastatic TNBC (mTNBC) is around 18 months [2,3]. As such, in spite of accounting for 15% of all breast cancers, TNBC represents a challenge for clinicians, with the necessity to develop new treatment strategies for this distinct subgroup [4]. 

At the molecular scale, a germline mutation in *BRCA1* or *BRCA2* (*gBRCA1/2*) is found in 15–20% of patients with TNBC [5,6]. The disruption of *BRCA1/2* genes leads to inefficient homologous recombination (HR) for double-strand break (DSB) processing, leading to the so-called homologous recombination deficiency (HRD) and subsequent specific features through genomic instability [7]. In the context of neoadjuvant chemotherapy, it has been shown that tumors with *BRCA1/2* mutations exhibit an increased sensitivity to platinum salts [8,9]. In the metastatic setting, current guidelines from the European Society for Medical Oncology (ESMO) recommend platinum-based chemotherapies in the context of *gBRCA1/2*-mutated TNBC [10]. Furthermore, in the context of *gBRCA1/2,* TNBCs exhibit higher sensitivity to Poly(ADP-ribose) polymerase inhibitors (PARPi) through synthetic lethality [7]. In recent years, several trials have shown the efficiency of PARPi’s in the context of *gBRCA1/*2, both in the metastatic and adjuvant settings [11,12,13]. Apart from the well-characterized *BRCA1/2*, several other causes leading to HRD have been described, such as the promoter hypermethylation of *BRCA1* or *RAD51C* or mutations in HR-related (HRR) genes, which are grouped under the concept of “*BRCAness*”, which represent any molecular alteration out of *BRCA1/2* that leads to an HRD phenotype [2,7]. In spite of their putative effect on HR function, it remains unclear if their presence in TNBC is associated with enhanced sensitivity to platinum-based chemotherapy. To address this question, we analyzed a specific panel of HRR genes and the promoter methylation of *BRCA1* or *RAD51C* of mTNBC from patients included in the ProfiLER-01 trial [14]. The objectives of the study were assessing the prevalence of alteration of HRR genes and their association with platinum-based chemotherapy.

## 2. Materials and Methods

### 2.1. Patients and Tumors

All mTNBC patients included in the ProfiLER-01 trial (NCT01774409) from March 2013 to July 2017 were analyzed in the present study. Characteristics of the ProfiLER-01 trial have been previously published [14]. This study was approved by Ethics Committee of Lyon Sud-Est IV in agreement with the Good Clinical Practice guidelines of the International Conference on Harmonization and the Declaration of Helsinki, and with relevant French and European laws and directives. All patients provided written informed consent. Treatment response was assessed by clinical and radiological evaluations every 2 to 3 months, and radiological evaluation was based on RECIST1.1 criteria [15]. Stable disease was defined by the response or absence of signs of progression after three months of treatment. Disease control included complete response, partial response and stable disease. Relapse free survival (RFS) was defined as the time from date of diagnosis of early breast cancer to the time of loco-regional relapse or metastatic relapse. Overall survival (OS) was defined as the time from the diagnosis of incurable cancer (metastatic or locally advanced without curative intent) to the date of death or end of follow-up or cutoff date. Median OS (OS) was defined as the time from beginning of first-line treatment to the date of death (for any cause) or end of follow-up or cutoff date. 

### 2.2. DNA Sequencing

DNA extraction was performed from Paraffin-embedded (FFPE) blocks (primary tumors or metastases). DNA-capture based next generation sequencing was performed using 50 ng of genomic DNA with a custom “HRR genes” panel of the following 19 selected genes: *ATM, BARD1, BRCA1, BRCA2, BRIP1, CDK12, CHEK1, CHEK2, FANCA, FANCD2, FANCL, MRE11, NBN, PALB2, PPP2R2A, RAD51B, RAD51C, RAD51D* and *RAD54L* (Sophia Genetics, Saint-Sulpice, Switzerland). The *TP53* gene was used as internal control. The human genome Hg19 (GRCh37.p5) was used as the reference genome. Fastq files were analyzed with the Sophia DDM Platform version 5.1.9 (Sophia Genetics, Saint-Sulpice, Switzerland). SIFT, Polyphen-2e Clinvar, NNSPLICE, MaxEnt, SSF and guidelines from the Association for Molecular Pathology, American Society of Clinical Oncology, and College of American Pathologists were used to classify each variant as “pathogenic”, “likely pathogenic”, “uncertain significance”, “likely benign”, or “benign”. Variants were considered as “pathogenic” or “likely pathogenic” if at least three databases or tools predicted it [16]. Some patients underwent genetic counseling and germline analysis based on their familial and clinical characteristics; if applicable, germline results were collected.

### 2.3. Methylation Assays

300 ng of genomic DNA were amplified with ddPCR after treatment with bisulfite, using the EpiGenteck kit (EpiGenteck Inc., Farmingdale, NY, USA) according to the manufacturer’s instructions. Previously published specific primers and probes for methylated *BRCA1* gene promoter analysis were used [17]. The *RAD51C* gene promoter was designed in-house. The *ACTB* gene was used as an internal amplification control as previously published [18]. Methylation percentage was defined as the ratio between levels of methylated *BRCA1* (or *RAD51C*) and *ACTB* genes.

### 2.4. HRD Groups

Patients were divided into three groups according to their molecular alterations. Group A included patients with *BRCA1/2* pathogenic or likely pathogenic mutations. Group B included patients without *BRCA1/2* mutations and with pathogenic or likely pathogenic mutations in at least one of the other genes from the “HRR genes” panel described upwards or with the promoter methylation of *BRCA1* or *RAD51C* genes. Group C included all of the remaining patients (i.e., without alterations detected).

### 2.5. Statistical Analyses

Between-group comparisons were performed using Fisher’s exact test for categorical data and a non-parametric Kruskall-Wallis test for continuous data. A *p*-value < 0.05 was considered significant. Survival data were performed using the Kaplan-Meier method. Missing data were censored to the date of latest news. SAS version 9.4 (SAS Institute, Cary, NC, USA) was used for all statistical analyses.

### 2.6. Study Endpoints

The primary endpoint was the description of the alteration of the genes of the HR pathway in mTNBC populations. Secondary endpoints were responses to treatment and the associations between molecular alterations and platinum efficacy.

## 3. Results

### 3.1. Patients’ and Treatments’ Characteristics

Fifty-nine patients from the ProFILER-01 trial exhibited mTNBC and were analyzed; seven patients were excluded due to the lack of genetic material or the poor quality of genetics samples. Fifty-two patients were divided into three groups with the following distribution: 7, 16 and 29 patients (groups A, B and C, respectively). Regarding clinical characteristics, all patients were females with median age at diagnosis significantly different (*p* = 0.006) with 37.1 years (range 27–40) in group A, 42.1 years (range 28–65) in group B and 48.2 years (range 34–65) in group C (Table 1).

Germline *BRCA* testing was performed in seven patients in group A, eight patients in group B, and 7 patients in group C. There were no significant differences between groups in terms of family history of cancer, tumor grades or number of metastatic sites. Forty-nine (94.2%) patients underwent surgery of the primary breast tumor; and 44 (84.6%) patients received chemotherapy in the (neo)-adjuvant setting (Table 2). Initially, local or locally advanced disease at diagnosis were as follows: five (71%), 14 (88%) and 25 (86%) patients in group A, B, and C, respectively. Upon neoadjuvant treatment, responses to therapy were evaluated in three, eight and nine patients in group A, B and C, respectively. One patient in group A had pathological complete response to neoadjuvant treatment, while two patients in group B and one patient in group C progressed on therapy. No difference was seen regarding PFS from localized therapy (*p* = 0.91). No patients received platinum-based treatment in this setting.

### 3.2. Molecular Alterations

From the 52 tumor samples, variant analysis, large rearrangement analysis of HRR genes and assessment of promoter methylation of *BRCA1* and *RAD51C* were performed on 52, 50 and 47 samples, respectively. Seven samples were discarded due to technical failure (*n* = 2) or insufficient tissue (*n* = 5). Tumor DNA was extracted from 40 primary tumors and 12 metastases. Among the seven patients in group A, six patients had *BRCA1* mutations and one had a *BRCA2* mutation; all of these mutations were also found constitutively. In group B, the mutated genes were as follows: *FANCA* (*n* = 2), *FANCL* (*n* = 1), *RAD51D* (*n* = 1; also detected as a germline mutation), *NBN* (*n* = 1) and *CHEK2* (*n* = 1). Promoters of *BRCA1* and *RAD51C* were found to be methylated in eight and two patients, respectively.

### 3.3. Response to Platinum-Based Chemotherapy

Among the 27 patients who received platinum-based chemotherapy, disease control rate at three months was 80% in group A, 55% in group B, and 18% in group C (*p* = 0.049; Figure 1a). When combining groups A and B versus group C (corresponding to presence or absence of molecular alterations, respectively), a statistically significant differential disease control remained (*p* = 0.047; Figure 1b).

Regarding survival associated with platinum-based chemotherapy, the median PFS were 5.3, 3.0, and 2.1 months (groups A, B and C, respectively), nevertheless without reaching a statistically significant difference (*p* = 0.36; Figure 2a). Similarly, median OS were as follows: 35.0, 27.2, and 32.4 months (groups A, B and C, respectively; *p* = 0.77; Figure 2b).

### 3.4. Exploratory Analysis at the Individual Scale

Individual responses are described in Figure 3. It is worthy of note that patient 50 carried a two-base germline mutation of *BRCA2* (c.3545_3546delTT) in exon 11, causing a frameshift variant and a premature termination codon. A second deletion c.3509_3547del was found on tumor analysis, which restored the reading frame and BRCA2 function.

## 4. Discussion

Beyond *BRCA1/2* mutations, this retrospective analysis shows that specific alterations of HR genes are associated with platinum sensitivity in mTNBC patients, namely *FANCL* mutation, *FANCA* mutation, *RAD51D* mutation and promoter methylation of *RAD51C*. In our study, we tested tumor DNA by NGS with a small HR-gene panel, as well as *BRCA1* and *RAD51C* promoter methylation. We found that 44% of TNBC tumors had HR gene alterations; and 30.7% had alterations in HRR genes other than *BRCA1/2*, in accordance with the literature. Indeed, HRD score evaluated with MyChoice**^®^** (Myriad Genetics Inc., Salt Lake City, UT, USA) was considered high for 41–71% of selected patients with early and metastatic cancer, but it remains unknown whether these patients may be eligible for PARPi TNBC [9,19,20,21]. In these studies, *BRCA1/2* mutations account for only 15–20% of tumors with high HRD score. In another study, the HRDetect mutational-signature-based algorithm score was high in 59% of patients with early TNBC [22]. 

Early TNBC with a high HRD score and especially *BRCA1/2* mutations showed higher pCR rates after neoadjuvant chemotherapy [19,20,23,24]. In the metastatic setting, the TNT trial failed to demonstrate the benefit of platinum-based chemotherapy for the whole cohort of TNBC patients, as well as for patients with high-HRD scores. Nevertheless, this trial showed that g*BRCA1/2* gene mutations were significantly associated with better tumor response to carboplatin than docetaxel treatment (68% versus 33%) [9]. In the present study, putative HRD tumors (with *BRCA 1/2* mutation or other HR gene alteration) presented a higher disease control rate than non-HRD patients after platinum-based chemotherapy. Patients with g*BRCA1* gene mutation responded to platinum-based chemotherapy, consistent with previous studies [9,25]. The only patient with a *gBRCA2* mutation who did not respond harbored a reverse mutation of the *BRCA2* gene. This resistance mechanism to platinum-based chemotherapy has been previously described [26]. In the context of putative HRD without *BRCA1/2* mutations, patients who responded to platinum-based chemotherapy harbored the following alterations: the promoter methylation of *RAD51C*, and mutations of the *RAD51D, FANCA* and *FANCL* genes. Each alteration represented less than 2% of our entire TNBC cohort, in concordance with existing data [27,28]. The methylation of *RAD51C* has been associated with “signature 3” like *BRCA1/2* mutated tumors and has been shown to exhibit the features of HRD tumors [22,27]. Regarding the 6 patients with promoter methylation of *BRCA1*, only one presented a stable disease with platinum-based chemotherapy in our cohort. Although this epimutation is considered as a marker of HRD, notably with its association with “signature 3”, its theragnostic impact has been controversial [27]. In the TNT trial, although a majority of tumors with *BRCA1* promoter methylation had a high HRD score (based on the MyChoice**^®^** assay), they did not exhibit a better response to carboplatin treatment [9]. The same absence of impact has been reported in the TBCR009 trial [25]. Nevertheless, this epimutation was associated with response to olaparib in untreated TNBC [29]. Several limits may explain this apparent discordance. Firstly, there is not an absolute correlation between promoter methylation and the absence of expression at the RNA level [17]. Secondly, we performed methylation assays on primary tumors for all patients but one; as such, based on the dynamic nature of methylation, we cannot exclude a different *BRCA1* promoter methylation status between the initial biopsy and the pre-platinum time points. Indeed, the reversion of *BRCA1* promoter methylation has already been reported, leading to HR pathway restoration [30]. 

Our study has several limits. Firstly, although it is an almost “real-life” trial regarding enrolled patients, the sample size is quite modest, at both the whole cohort and platinum-challenged patient’s scales. This may have led to the absence of statistical significance regarding the different PFS observed after platinum-based chemotherapy. Secondly, as previously discussed, the vast majority of molecular analyses were performed on tissues obtained from initial biopsies; as such, the molecular alterations observed do not necessarily perfectly reflect the actual status upon platinum-challenge. Nevertheless, assessing the predictive value of initial biomarkers allowed us to decipher more robust ones (i.e., less sensitive to evolutionary fluctuations); furthermore, iterative biopsies with multiple molecular assays are not necessarily compatible with real-life practice. Thirdly, it is now well-known that HRD status may be assessed through diverse companion diagnostic assays; they nevertheless do not overlap perfectly [7]. The MyChoice**^®^** assay has been shown to represent the current gold standard in ovarian cancer as a prerequisite to PARPi challenge [31]. As such, it would have been interesting to analyze the potent correlation between the molecular alterations observed in our study and the associated HRD score. Regarding perspectives, PARPi have already demonstrated their interest in breast tumors with *gBRCA1/2* [11,12]. Results from the phase II RUBY trial, which assessed rucaparib in patients with an HRD status evaluated with the Foundation Medicine**^®^** T5 assay (Foundation Medicine, Cambridge, MA, USA) in mTNBC showed encouraging data regarding larger populations than *gBRCA1/2* patients [32].

## 5. Conclusions

Our present study showed that 30% of TNBC tumors had at least one mutation in our HRR-gene panel (*BRCA1/2* genes excluded), and these specific alterations were associated with an objective response to platinum-based chemotherapy. As such, a more comprehensive approach should be considered when assessing the HRD status for TNBC patients, with the possible effect of enlarging the selection of patients that could benefit from platinum-based treatments. Further studies are needed regarding the impact of HRR-genes in mTNBC and their predictive impact regarding sensitivity to PARPi.

## Figures and Tables

**Figure 1 jpm-12-01595-f001:**
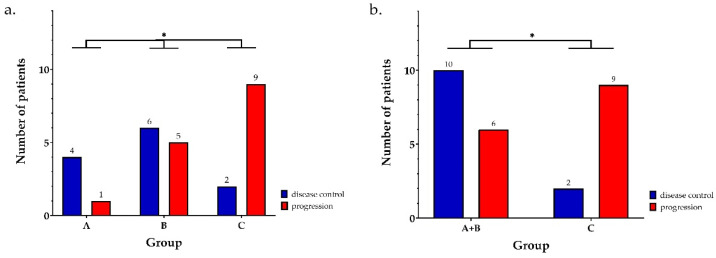
Response to platinum-based therapy according to molecular status phenotype. Responses are given for each group taken separately (**a**), or by comparing patients with or without molecular alteration (groups A + B versus group C respectively) (**b**). * represents differences with statistical significance (*p* < 0.05).

**Figure 2 jpm-12-01595-f002:**
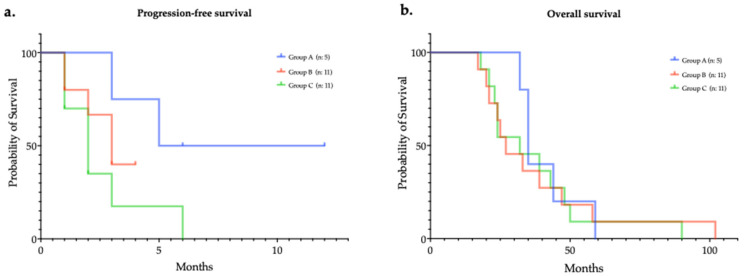
Progression-free (**a**) and overall (**b**) survivals of patients challenged with platinum-based chemotherapy, according to molecular status.

**Figure 3 jpm-12-01595-f003:**
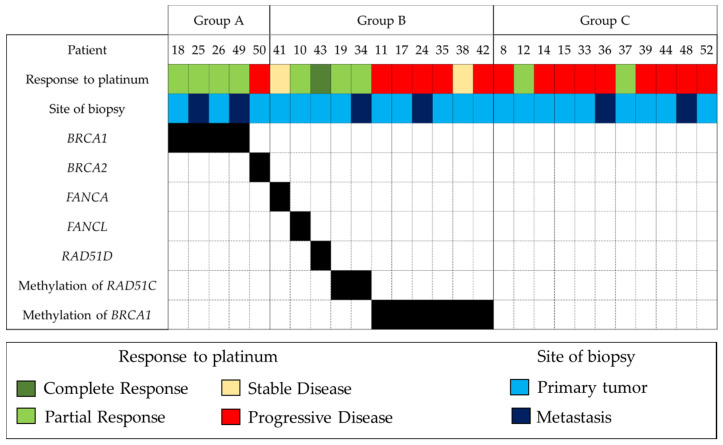
Site of biopsy, molecular status and radiological response for each patient challenged with platinum-based chemotherapy. A black square indicates the presence of a given molecular alteration.

**Table 1 jpm-12-01595-t001:** Patient’s characteristics.

	Group A *n* = 7	Group B *n* = 16	Group C *n* = 29	All *n* = 52	*p-*Value **
**Age at diagnosis**					
Median (min–max)	37.1 (27–40)	42.1 (28–65)	48.2 (34–65)	42.1 (27–65)	0.006
**Personal history of breast cancer** ***n*, (%)**	0	0	4 (14%)	4 (14%)	
**Family history of cancer ***					
1st and/or 2nd degree	4 (57%)	7 (44%)	7 (25%)	18 (35%)	0.16
No history	3 (43%)	9 (56%)	22 (75%)	34 (65%)	
*gBRCA1/2* testing *n* (%)	7 (100%)	8 (50%)	7 (24%)	22 (42%)	
**Histology, *n* (%)**					
NOS	7 (100%)	13 (81%)	25 (86%)	45 (87%)	
Lobular	0	0	1 (4%)	1 (2%)	
other	0	3 (19%)	3 (10%)	6 (11%)	
**Grade, *n* (%)**					
1	0	0	2 (7%)	2 (4%)	
2	0	3 (20%)	7 (24%)	10 (20%)	
3	6 (100%)	12 (80%)	20 (69%)	38 (76%)	
Unknown	1	1	0	2	
**Number of metastatic sites**					
1	1 (14%)	5 (31.3%)	12 (41.4%)	18 (34.6%)	
2–3	6 (86%)	11 (68.8%)	13 (44.8%)	30 (57.7%)	
>3	0	0	4 (13.8%)	4 (7.7%)	*p* = 0.21
**Sites of metastasis, *n* (%)**					
Bones	1 (14%)	5 (31%)	10 (34%)	16 (31%)	
CNS	0	2 (12%)	2 (7%)	4 (8%)	
Skin	2 (29%)	5 (31%)	5 (17%)	12 (23%)	
Liver	0	3 (19%)	8 (28%)	11 (21%)	
Lung	5 (71%)	7 (44%)	14 (50%)	26 (51%)	
Regional lymph nodes	5 (71%)	6 (37%)	17 (58%)	28 (54%)	

Abbreviations: *gBRCA1/2*: germline mutation of *BRCA1/2*; NOS: not otherwise specified. * Ovary, breast, prostate or uterus. ** Kruskal-Wallis test or Fisher exact test.

**Table 2 jpm-12-01595-t002:** Treatment’s Characteristics.

	Group A *n* = 7	Group B *n* = 16	Group C *n* = 29	All *n* = 52	*p*-Value *
**Disease stage at diagnosis**					
Local/Locoregional	5 (71%)	14 (88%)	25 (86%)	44 (85%)	
Metastatic	2 (29%)	2 (12%)	4 (14%)	8 (15%)	
**Neo/adjuvant chemotherapy**	*n* = 5 (71%)	*n* = 14 (88%)	*n* = 25 (86%)	*n* = 44 (85%)	
Cyclophosphamide	5 (100%)	13 (92%)	25 (100%)	43 (97%)	
Anthracycline	5 (100%)	11 (78%)	24 (96%)	40 (90%)	
Taxane	5 (100%)	13 (92%)	24 (96%)	42 (95%)	
**Breast surgery**	7 (100%)	14 (87%)	28 (97%)	49 (94%)	
Nodes involvement	1 (14%)	11 (68%)	17 (59%)	29 (55%)	
**Response to neoadjuvant therapy**	*n* = 3	*n* = 8	*n* = 9	*n* = 20	
pCR	1 (33%)	0	0	1 (5%)	
PR/SD	2 (67%)	6 (75%)	8 (89%)	16 (80%)	
PD on therapy	0	2 (25%)	1 (11%)	3 (15%)	
**Adjuvant RT**	5 (71%)	14 (88%)	21 (72%)	40 (76%)	
**RFS from localized therapy (months)** Median (min–max)	16.0 (11–17)	14.3 (7–53)	13.6 (3–202)	14.2 (3–202)	*p* = 0.91
**Number of lines in metastatic setting** Median (min–max)	4 (2–6)	6 (1–9)	3 (1–9)	4 (1–9)	
**OS (months)** Median (min–max)	19.2 (8.0–44.0)	23.3 (14.5–31.2)	16.6 (10.2–28.4)	17.9 (8.0–44.0)	*p* = 0.86

Abbreviations: CNS: Central Nervous System ET: endocrine therapy; gBRCA1/2: germline BRCA1/2 mutation; NOS: not otherwise specified; OS: overall survival; pCR: pathological complete response; PD: progression disease; PR: partial response; RFS: Response Free Survival; RT: Radiation therapy; SD: Stable disease. * Fisher exact test or Kruskall-Wallis test.

## Data Availability

Datasets are available upon request to corresponding author.

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
