# Peer review of "Alterations in Homologous Recombination-Related Genes and Distinct Platinum Response in Metastatic Triple-Negative Breast Cancers: A Subgroup Analysis of the ProfiLER-01 Trial"

_jpm, 2022, doi:10.3390/jpm12101595_

Round 1

Reviewer 1 Report

In this manuscript the authors performed a retrospective clinical analysis investigating whether non-BRAC mutation-dependent homologous recombination deficiency (HRD) enhances sensitivity to platinum-regimen chemotherapy. This is a well written manuscript presenting clinically relevant data that could significantly improve the survival of patients with triple-negative breast cancer (TNBC). I especially appreciate the discussion where the authors do a nice job of explaining how their data fit into the larger literature on the topic as well as going into the limitations of this study. A few minor edits will help to round out this manuscript:

1.    Due to the clinical nature of the results, the addition of a few details to the Introduction would help to highlight the significance of the findings.

a.     The authors discuss the concept of ‘BRCAness’ in the introduction (line 70), but they do not discuss how this ‘BRCAness’ is addressed in the clinic. For instance, is the ‘BRCAness’ of TNBC tumors tested in patients?

b.    Since the major focus of this paper is the sensitivity of TNBC tumors to platinum therapies, it would be useful for the authors to briefly discuss the usual treatment regime for TNBC patients and when platinum drugs are introduced into patients’ treatment plans.

2.    There is a spelling error at the beginning of the sentence in line 58.

3.    In the Materials and Methods, Patients and Tumors section the authors define Median OS as including time to death and “end of follow-up or cutoff date” (line 94). This makes it sound like survival times may be included in this dataset for patients that may have survived beyond the “follow-up” date but were given a premature survival date. Please clarify this and describe why follow-up/cutoff dates were included.

4.    Figure 1 graphs could be clarified a bit.

a.     Label the y-axis.

b.    In the legend define what the ‘Control’ data represents. I assume that these are the patients that did not progress, but that is not made clear in the figure or the legend.

5.    Figure 2 is not referenced in the text of the Results section.

Author Response

The authors sincerely thank the reviewer for the inputs. Please find underneath the response to all of the other comments.

In this manuscript the authors performed a retrospective clinical analysis investigating whether non-BRAC mutation-dependent homologous recombination deficiency (HRD) enhances sensitivity to platinum-regimen chemotherapy. This is a well written manuscript presenting clinically relevant data that could significantly improve the survival of patients with triple-negative breast cancer (TNBC). I especially appreciate the discussion where the authors do a nice job of explaining how their data fit into the larger literature on the topic as well as going into the limitations of this study. A few minor edits will help to round out this manuscript:

  1. Due to the clinical nature of the results, the addition of a few details to the Introduction would help to highlight the significance of the findings.

Thanks for this comment, we actually added “As such, in spite of accounting for 15% of all breast cancers TNBC represent a challenge for clinicians, with the necessity to develop new treatment strategies for this distinct subgroup.”

  1. The authors discuss the concept of ‘BRCAness’ in the introduction (line 70), but they do not discuss how this ‘BRCAness’ is addressed in the clinic. For instance, is the ‘BRCAness’ of TNBC tumors tested in patients?

Thanks for this comment, we actually added this sentence in lines 72-73: “Furthermore, these types of alterations are not currently assessed in clinical routine.”

  1. Since the major focus of this paper is the sensitivity of TNBC tumors to platinum therapies, it would be useful for the authors to briefly discuss the usual treatment regime for TNBC patients and when platinum drugs are introduced into patients’ treatment plans.

Thanks for this comment, we actually added the sentence “In the metastatic setting, current guidelines from the European Society for Medical Oncol-ogy (ESMO) recommend platinum-based chemotherapies in the context of gBR-CA1/2-mutated TNBC [10]”.

  1. There is a spelling error at the beginning of the sentence in line 58.

Thanks for this comment, we changed it towards “AT the …”

  1. In the Materials and Methods, Patients and Tumors section the authors define Median OS as including time to death and “end of follow-up or cutoff date” (line 94). This makes it sound like survival times may be included in this dataset for patients that may have survived beyond the “follow-up” date but were given a premature survival date. Please clarify this and describe why follow-up/cutoff dates were included.

Thanks for this comment, in order to avoid misunderstaninding, we changed it towards « Overall survival (OS) was defined as the time from the diagnosis of incurable cancer (metastatic or locally advanced without curative intent) to the date of death or end of fol-low-up or cutoff date. Median OS (OS) was defined as the time from beginning of first-line treatment to the date of death (for any cause) or end of follow-up or cutoff date. »

  1. Figure 1 graphs could be clarified a bit.

  1. Label the y-axis.
  2. In the legend define what the ‘Control’ data represents. I assume that these are the patients that did not progress, but that is not made clear in the figure or the legend.

Thanks for this comment, we actually modified the figure in order to add: colors, higher resolution, y-label axis, modification of “control” towards “disease control”

  1. Figure 2 is not referenced in the text of the Results section.

Thanks for this comment, there was actually a problem in formatting the article, as the following sentences “Regarding survivals associated with platinum-based chemotherapy, median PFS were 5.3, 3.0, and 2.1 months (groups A, B and C, respectively), nevertheless without reaching statistically significant difference (p=0.36; Figure 2a.). Similarly, median OS were as fol-lows: 35.0, 27.2, and 32.4 months (groups A, B and C, respectively; p=0.77 ; Figure 2b.).” were included in the legend of figure 2. We changed it.

Reviewer 2 Report

I would like to congratulate the authors on the quality of their study as well as its presentation. The study is of high value to the field and concerns mutations in HRD genes and promoter methylation of specific genes in the sensitivity of TNBC cancers to treatment using platinum-based chemotherapy. The authors conclude that mutations in BRCA1/2 as well as at least one if the HRD genes they assayed increase the sensitivity of these tumors t platinum based therapy as does methylation of the promoters for BRCA1/2 and RAD

I only have a few minor comments

1)      The series labels for figure 1 are very small. I would suggest these be made clearer by making them bigger

2)      Throughout the paper “the” is used either in the wrong place or is missing. For instance, in the abstract disease control rate should be the disease control rate in the introduction in ProfiLER-01 should be in the ProfiLER-01 in the methods section Association should be The Association. In the results section poor quality should be the poor quality, Alternatively in the results groips are described as the group A. It dhould be just group A

3)      When presenting the median age of the patients at the beginning of the results section the first range is missing a hyphen. In addition the use of the square brackets means these numbers could be mistaken to be references, I would suggest using round brackets

4)      In the abstract Separated into 3 groups

5)      In the Introduction some minor errors include

TNBC represents

 At the molecular

 In the last few/ in recent years, several

6)      In the methods section some minor errirs include

date of diagnostic of should be date of diagnosis of

 performed over Paraffin should be performed from Paraffin

 Variant were should be variants were

  for continues data .should be for continuous data

 with platinum efficacy should be treated effectively with platinum

7)       In the results section some minor errors include

had a BRCA2 mutation; noteworthy, these- I am not sure what the authors wanted to say here but the maybe had a noteworthy a BRCA2 mutation

8)      In the Discussion some minor errors include

had other HR gene alterations than BRCA1/2, should be had HR gene alterations other than BRCA1/2,

The references [23,24,27,28] are in bold

as well for patients should be as well as for patients

gBRCA1/2 genes mutations should be gBRCA1/2 gene mutations

consistently with previous studies should be consistent with previous studies

had an high HRD should be had a high HRD

patients enrolled should be enrolled patients

allows to decipher more robust ones should be allows us to decipher more robust ones

Author Response

I would like to congratulate the authors on the quality of their study as well as its presentation. The study is of high value to the field and concerns mutations in HRD genes and promoter methylation of specific genes in the sensitivity of TNBC cancers to treatment using platinum-based chemotherapy. The authors conclude that mutations in BRCA1/2 as well as at least one if the HRD genes they assayed increase the sensitivity of these tumors t platinum based therapy as does methylation of the promoters for BRCA1/2 and RAD

The authors sincerely thank the reviewer for the supportive comments and specific inputs, allowing to improve the overall quality of our manuscript.

Please find underneath the response to all of the other comments.

I only have a few minor comments

  • The series labels for figure 1 are very small. I would suggest these be made clearer by making them bigger

Thanks for this comment, we actually modified the figure 1 in order to add: colors, higher resolution, y-label axis (as we did for figure 3).

  • Throughout the paper “the” is used either in the wrong place or is missing. For instance, in the abstract disease control rate should be thedisease control rate in the introduction in ProfiLER-01 should be in the ProfiLER-01 in the methods section Association should be The Association. In the results section poor quality should be the poor quality, Alternatively in the results groips are described as the group A. It dhould be just group A

Thanks for this input, we changed it all along the manuscript.

  • When presenting the median age of the patients at the beginning of the results section the first range is missing a hyphen. In addition the use of the square brackets means these numbers could be mistaken to be references, I would suggest using round brackets

Thanks for this relevant input, we changed it.

In the abstract Separated into 3 groups

Thanks for this input, we modified it.

5)      In the Introduction some minor errors include

TNBC represents

 At the molecular

 In the last few/ in recent years, several

6)      In the methods section some minor errirs include

date of diagnostic of should be date of diagnosis of

 performed over Paraffin should be performed from Paraffin

 Variant were should be variants were

  for continues data .should be for continuous data

 with platinum efficacy should be treated effectively with platinum

7)       In the results section some minor errors include

had a BRCA2 mutation; noteworthy, these- I am not sure what the authors wanted to say here but the maybe had a noteworthy a BRCA2 mutation

8)      In the Discussion some minor errors include

had other HR gene alterations than BRCA1/2, should be had HR gene alterations other than BRCA1/2,

The references [23,24,27,28] are in bold

as well for patients should be as well as for patients

gBRCA1/2 genes mutations should be gBRCA1/2 gene mutations

consistently with previous studies should be consistent with previous studies

had an high HRD should be had a high HRD

patients enrolled should be enrolled patients

allows to decipher more robust ones should be allows us to decipher more robust ones

Thanks for these inputs, we modified the manuscript and modified all the underscored minor errors.

Reviewer 3 Report

A very interesting study 

Abstract: Well written and includes the key results

Introduction: Concise and introduced the subjects of the study

Result:

Line 139:  (supplementary Figure 1: flowchart) can be removed and add the details in the text.

For all figures, the first letter after the word Figure x should be in capital, eg: Figure 2. Progression-free (2a.) 

Line 169: Appendix A can be removed and the reason for the removal of the samples can be included in the text.

References: Almost 40% of the references were > 5 years old. Please re-check and ensure if there are recent references that can be used.

Author Response

A very interesting study 

Abstract: Well written and includes the key results

Introduction: Concise and introduced the subjects of the study

The authors sincerely thank the reviewer for the supportive comments and specific inputs, allowing to improve the overall quality of our manuscript.

Result:

Line 139:  (supplementary Figure 1: flowchart) can be removed and add the details in the text.

Thanks for this remark, we removed the supplementary figure.

For all figures, the first letter after the word Figure x should be in capital, eg: Figure 2. Progression-free (2a.) 

Thanks for this remark, we added capitals where necessary.

Line 169: Appendix A can be removed and the reason for the removal of the samples can be included in the text.

Thanks for this remark, we removed the supplementary figure and added detail in the texte.

References: Almost 40% of the references were > 5 years old. Please re-check and ensure if there are recent references that can be used.

Thanks for this remark, except from clinical trials and original articles required for protocol consideraitons, we modified all the references in order to reach an up to date bibliography.